# Greener Pretreatment Approaches for the Valorisation of Natural Fibre Biomass into Bioproducts

**DOI:** 10.3390/polym13172971

**Published:** 2021-08-31

**Authors:** Mohd Nor Faiz Norrrahim, Muhammad Roslim Muhammad Huzaifah, Mohammed Abdillah Ahmad Farid, Siti Shazra Shazleen, Muhammad Syukri Mohamad Misenan, Tengku Arisyah Tengku Yasim-Anuar, Jesuarockiam Naveen, Norizan Mohd Nurazzi, Mohd Saiful Asmal Rani, Mohd Idham Hakimi, Rushdan Ahmad Ilyas, Mohd Azwan Jenol

**Affiliations:** 1Research Center for Chemical Defence, Universiti Pertahanan Nasional Malaysia, Kem Sungai Besi, Kuala Lumpur 57000, Malaysia; 2Department of Bioprocess Technology, Faculty of Biotechnology and Biomolecular Sciences, Universiti Putra Malaysia UPM, Serdang 43400, Malaysia; idhamhakimi@ymail.com (M.I.H.); azwan.jenol@gmail.com (M.A.J.); 3Faculty of Agricultural Science and Forestry, Bintulu Campus, Universiti Putra Malaysia, Bintulu Sarawak 97000, Malaysia; 4Laboratory of Biopolymer and Derivatives, Institute of Tropical Forestry and Forest Products (INTROP), Universiti Putra Malaysia UPM, Serdang 43400, Malaysia; shazra.shazleen@yahoo.com; 5Department of Chemistry, College of Arts and Science, Yildiz Technical University, Davutpasa Campus, Esenler, Istanbul 34220, Turkey; syukrimisenan@gmail.com; 6Nextgreen Pulp & Paper Sdn. Bhd., Menara LGB, Jalan Wan Kadir, Taman Tun Dr. Ismail, Kuala Lumpur 60000, Malaysia; tengkuarisyah@gmail.com; 7School of Mechanical Engineering, Vellore Institute of Technology, Vellore 632014, India; gandhi.naveen66@gmail.com; 8Center for Defence Foundation Studies, Universiti Pertahanan Nasional Malaysia, Kem Perdana Sungai Besi, Kuala Lumpur 57000, Malaysia; 9School of Materials and Mineral Resources Engineering, Engineering Campus, Universiti Sains Malaysia, Nibong Tebal 14300, Malaysia; saifulasmal@gmail.com; 10School of Chemical and Energy Engineering, Faculty of Engineering, Universiti Teknologi Malaysia UTM, Johor Bahru 81310, Malaysia

**Keywords:** non-chemical pretreatment, lignocellulosic biomass, bioproducts

## Abstract

The utilization of lignocellulosic biomass in various applications has a promising potential as advanced technology progresses due to its renowned advantages as cheap and abundant feedstock. The main drawback in the utilization of this type of biomass is the essential requirement for the pretreatment process. The most common pretreatment process applied is chemical pretreatment. However, it is a non-eco-friendly process. Therefore, this review aims to bring into light several greener pretreatment processes as an alternative approach for the current chemical pretreatment. The main processes for each physical and biological pretreatment process are reviewed and highlighted. Additionally, recent advances in the effect of different non-chemical pretreatment approaches for the natural fibres are also critically discussed with a focus on bioproducts conversion.

## 1. Introduction

Many industries currently produce many tons of agro-industrial wastes. However, direct utilization of lignocellulosic biomass as a feedstock for bioproducts is challenging due to their complex structure (as represented in Figure 1). A variety of useful components, including sugars, protein, lipids, cellulose, and lignin, are present in natural fibres. The major issue that limits their utilization is, however, the tight bonding within their components [1]. Cross-linking of polysaccharides and lignin occurs through ester and ether bonds, while microfibrils produced by cellulose, hemicellulose, and lignin aid in the stability of plant cell wall structure [2,3]. These strong cross-linking connections exist between the components of the plant cell wall that act as a barrier to its disintegration.

Pretreatment helps to fractionate biomass prior to further processes, making it simpler to handle in the process [4,5,6]. It enables biomass hydrolysis and makes building blocks for biobased products, fuels, and chemicals. It is often the initial stage of the biorefining process and enables the following steps such as enzymatic hydrolysis and fermentation to be carried out more quickly, effectively, and economically [7]. The pretreatment method used is entirely dependent on the targeted application. Numerous pretreatment methods are mainly developed to effectively separate these interconnected components in order to get the most advantages from the lignocellulosic biomass’s constituents.

Pretreatment of natural fibres is not as straightforward as it may seem. In fact, it is the second most expensive procedure after the installation of a power generator. Hydrogen bond disruption, cross-linked matrix disruption, as well as increased porosity and surface area, are the three objectives that a good pretreatment technique accomplishes in crystalline cellulose. Additionally, the result of pretreatment varies attributed to the different ratios of cell wall components [8]. More criteria to take into consideration for efficient and economically feasible pretreatment process include less chemical usage, prevention of hemicellulose and cellulose from denaturation, minimum energy demand, low price, and the capacity to reduce size.

Biomass recalcitrance is a term used for the ability of natural fibres to resist chemical and biological degradation. While there are many components involved in the recalcitrance of lignocellulosic biomass, the crystalline structure of cellulose, the degree of lignification, accessible surface area (porosity), the structural heterogeneity, and complexity of cell-wall are primary causes [9,10]. As a consequence of breaking the resistant structure of lignocellulose, it causes lignin sheath, hemicellulose, and crystallinity to all be degraded, as well as casuing a decrease in cellulose’s degree of polymerization [11]. 

Depending on the types of natural fibres employed, the preference for the pretreatment method varies according to the composition of cellulose, hemicellulose, and lignin. Figure 2 depicts the general differences between the many common approaches which come under the four categories of physical, chemical, biological, and combination pretreatment [4]. While some of these methods have successfully transitioned from a research platform to an industrial stage, there are many hurdles, and one of the greatest is the requirement for highly toxic waste generation and high-energy inputs. From here, a serious issue that must be addressed is the lack of green and cost-effective solutions. Nevertheless, it has only lately garnered significant attention as a potential solution to the problem by focusing on the employment of non-chemical pretreatment. This could be reflected by the increment in article publications that reviewed lignocellulosic fibre pretreatment via individual greener approach as highlighted in Table 1 indicating that this topic is increasingly well-known owing to environmental concerns. The development of technology that maximises the use of raw resources, reduces waste, and avoids the use of poisonous and hazardous compounds is critical to accomplishing this objective. However, a review of all greener pretreatment approaches for lignocellulosic biomass is missing in the current literature.

Hence, the green pretreatment approaches for lignocellulosic biomass such as physical, biological, and combination methods, as well as their impact on the separation of the complex components of different lignocellulosic sources, are reviewed in more detail in the next sections.

## 2. Physical Pretreatment

The physical pretreatment allows increasing the specific surface area of the fibres via mechanical comminution. It also contributes to reduce the crystallinity of the natural fibres and enhance their digestibility. The physical pretreatment usually does not affect the chemical composition of natural fibres. Physical pretreatment can be conducted by using milling, extrusion and ultrasound. Physical pretreatment is often an essential step prior to or following chemical or biochemical processing. However, the information on the mechanism of how physical pretreatment modifies the structures of the fibre is still limited. 

There are some drawbacks of physical pretreatment that need to be considered. Physical pretreatment lacks the ability to remove the lignin and hemicellulose which limits the enzymes’ access to cellulose. Besides that, physical pretreatment requires high energy consumption which limits its large-scale implementation and environmental safety concerns.

### 2.1. Mechanical Extrusion

Mechanical extrusion is one of the most conventional methods of pretreatment [17]. In this pretreatment, the fibres are subjected to a heating process (>300 °C) under shear mixing. Due to the combined effects of high temperatures that are maintained in the barrel and the shearing force generated by the rotating screw blades, the amorphous and crystalline cellulose matrix in the biomass residues is disrupted. Besides that, extrusion requires a significant amount of high energy, making it a cost-intensive method and difficult to scale up for industrial purposes [17].

Temperature and screw speed of extrusion are the main important factors. Karunanithy and Muthukumarappan [18] studied the effect of these factors on the pretreatment of corn cobs. When pretreatment was carried out at different temperatures (25, 50, 75, 100, and 125 °C) and different screw speeds (25, 50, 75, 100, and 125 rpm), maximum concentration sugars were obtained at 75 rpm and 125 °C using cellulase and β-glucosidase in the ratio of 1:4, which were nearly 2.0 times higher than the controls. 

### 2.2. Milling

Mechanical milling is used to reduce the crystallinity of cellulose. It can reduce the size of fibre up to 0.2 mm. However, studies found that further reduction of biomass particles below 0.4 mm has no significant effect on the rate and yield of hydrolysis [17]. The type of milling and milling duration are important factors that influence the milling process. These factors can greatly affect the specific surface area, the final degree of polymerization, and a net reduction in cellulose crystallinity. 

Wet disk milling has been a popular mechanical pretreatment due to its low energy consumption as compared to other milling processes. Disk milling enhances cellulose hydrolysis by producing fibres and more effective as compared to hammer milling which produces finer bundles [19]. Hideno et al. [20] compared the effect of wet disk milling and conventional ball milling pretreatment method over rice straw. The optimal conditions obtained were 60 min of milling time in case of dry ball milling while 10 repeated milling operations were required in case of wet disk milling. 

### 2.3. Ultrasound

Ultrasound is relatively a new technique used for the pretreatment of fibres [17]. Ultrasound waves affect the physical, chemical, and morphological properties of fibres. Ultrasound treatment leads to the formation of small cavitation bubbles. These bubbles can rupture the cellulose and hemicellulose fractions. The ultrasonic field is influenced by ultrasonic frequency and duration, reactor geometry, and types of solvent used. Besides that, fibres characteristics and reactor configuration also influence the pretreatment [21]. 

The power and duration of ultrasound are important to be optimised depending on the fibres and slurry characteristics. This is important to meet the pretreatment objectives. Duration of ultrasound pretreatment has maximum effect on pretreatment of fibres. Besides that, a higher ultrasound power level has an adverse effect on the pretreatment. It can lead to the formation of bubbles near the tip of the ultrasound transducer which hinders the transfer of energy to the liquid medium [22]. 

## 3. Biological Pretreatment 

Retting is a biological process in which enzymatic activity removes non-cellulosic components connected to the fibre bundle, resulting in detached cellulosic fibres. The dew retting uses anaerobic bacteria fermentation and fungal colonization to produce enzymes that hydrolyse fibre-binding components on fibre bundles. *Clostridium* sp. is an anaerobic bacterium commonly found in lakes, rivers, and ponds. Plant stems were cut and equally scattered in the fields during the dew retting process, where bacteria, sunlight, atmospheric air and dew caused the disintegration of stem cellular tissues and sticky compounds that encircled the fibres [23]. For the dew retting procedure to enhance fungal colonization, locations with a warm day and heavy might dew are recommended.

Bleuze et al. [24] investigated the flax fibre’s modifications during the dew retting process. Microbial colonization can be affected the chemical compositions of cell walls. After seven days, fungal hyphae and parenchyma were found on the epidermis and around fibre bundles, respectively. After the retting process (42 days), signs of parenchyma deterioration and fibre bundle decohesion revealed microbial infestation at the stem’s inner core.

Fila et al. [25] found 23 different varieties of dew-retting agent fungi in Southern Europe. All *Aspergillus* and *Penicillium* strains yield high-quality retted flax fibres, according to the researchers. Besides that, under field conditions, Repeckien and Jankauskiene [26] investigated the effects of fungal complexes on flax dew-retting acceleration. *Cladosporium* species variations with high colonization rates (25–29%) have been identified as a good fungus for fibre separation. Most fungi survived on flax fed with fungal complex N-3, which contained six different fungal strains.

On a commercial scale, Jankauskiene et al. [27] optimised the dew retting method. Two fungal combinations were created and put to straw after the swath was pulled and returned. Furthermore, after spraying *Cladosporium herbarum* suspension during fibre harvesting, extremely high fibre separation was found.

### Bacterial and Fungi Interaction

Fungi colonization is thought to be the most important enzymatic active mechanism for dew retting. Recent research has focused on the interplay of the bacterial and fungal communities during dew retting. The association between the chemical contents of hemp fibres and microbial population fluctuation during the retting process was investigated by Liu et al. [28]. In the first seven days, fungal colonization was discovered with very little bacteria. After 20 days, there was a gradually risen in bacterial attachments on the fibre surface, with fewer fungal hyphae. The area with the highest bacterial concentration was found to severely deteriorate. The phylogenetic tree for the bacterial and fungal population in dew-retting hemp fibres is shown in Figure 3. While Table 2 shows ultrastructural changes in hemp stems and fibres as a result of microbial activity during the retting process.

## 4. Combination Pretreatment

It can be noticed from the green pretreatment techniques applied to pretreat the lignocellulosic biomass reviewed in the previous section that, while each pretreatment method makes a significant contribution, no single pretreatment approach yields efficient results without its own inherent limitations. Therefore, the combined pretreatment strategies could minimise the drawbacks while still achieving the intended result.

### 4.1. Physiochemical Pretreatment

Physiochemical pretreatment could be achieved by temperature elevation and irradiation in the processing of lignocellulosic material. Physiochemical pretreatment by steam such as superheated steam, hydrothermal and steam explosion is the most common pretreatments applied on natural fibre for several purposes. Physiochemical pretreatment is usually applied to remove the hemicellulose and lignin from the natural fibres [30].

#### 4.1.1. Superheated Steam

Pretreatment of fibres by superheated steam is gaining interest recently, as this pretreatment is considered as an environmentally friendly technique to remove hemicellulose. This could be a great alternative to chemical pretreatment in order to isolate the cellulose. Superheated steam is believed as the most economical pretreatment as compared to the other physical pretreatments as discussed before. 

Superheated steam is unsaturated (dry) steam generated by the addition of heat to saturated (wet) steam [31]. It has several advantages such as improved energy efficiency, higher drying rate, being conducted at atmospheric pressure and reduced environmental impact when condensate is reused [32,33]. Saturated steam cannot be superheated when it is in contact with water which is also heated, and condensation of superheated steam cannot occur without being reduced to the temperature of saturated steam. It has a high heat transfer coefficient, enabling rapid and uniform heating. Drying rates with superheated steam are faster than those with conventional hot air. Steam in a dried state or superheated steam is assumed to behave like a perfect gas. Although superheated steam is considered a perfect gas, it possesses properties like those of gases namely pressure, volume, temperature, internal energy, enthalpy and entropy. The pressure, volume, and temperature of steam as a vapour is not connected by any simple relationship such as is expressed by the characteristic equation for a perfect gas. Figure 4 shows the schematic diagram of superheated steam pretreatment. The saturated steam was generated in the boiler. The saturated steam produced was further heated by a super-heater to produce superheated steam. Then, the superheated steam was subjected to the fibres.

Superheated steam has been managed to alter the chemical composition of natural fibres. It has been proven that superheated steam pretreatment managed to remove high amount of hemicellulose from the lignocellulose fibres [34,35,36,37,38,39,40]. According to Warid et al. [40], superheated steam pretreatment on oil palm biomass at higher temperature and shorter time managed to remove a high amount of hemicellulose while maintaining the cellulose composition as compared to the method reported by Norrrahim et al. [39]. It was found that oil palm mesocarp fibre pretreated at 260 °C/30 min managed to remove hemicellulose of 68%, while cellulose degradation is maintained below 5%. Besides that, superheated steam was also able to remove silica bodies from the fibres where the presence of silica bodies increases the difficulty in grinding the fibre and causes abrasive wear and screw damage [32].

#### 4.1.2. Hydrothermal

Hydrothermal treatment is another pretreatment that has been proven to effectively remove impurities such as hemicellulose, lignin, and silica from lignocellulosic biomass. This treatment is being widely used in industry, owing to its low cost of production, high effectiveness in removing impurities without affecting the cellulose structure, disorganizing hydrogen bonds, swelling of the lignocellulosic biomass, as well as minimum requirements of preparation and handling [41,42]. In contrast to the superheated steam system that uses steam as the main mechanism, hydrothermal pretreatment only relies on water that will be subjected to a high temperature during the whole processing [43]. This treatment is also considered as an autohydrolysis of lignocellulosic linkages, with the presence of hydronium ions (H_3_O^+^) generated from water and acetic groups released from hemicellulose. The hydronium ions (H_3_O^+^) will act as a catalyst to break down and loosen the lignocellulosic structure [41,44]. This then will improve the effectiveness of further treatments such as enzymatic hydrolysis for biosugar production [43] and anaerobic digestion for biomethane production [45]. 

Numerous studies have reported the effectiveness of this treatment in reducing impurities, especially at a very high temperature. Zhang et al. [46] studied the effects of different hydrothermal temperatures which were 170, 190, and 210 °C at 20 min pretreatment time on corn stover. This study reported a drastic reduction in hemicellulose with an increase in hydrothermal temperature. In fact, no content of hemicellulose was detected and almost 125% of lignin was removed after hydrothermal treatment at 210 °C. Similarly, Phuttaro et al. [47] also reported the same trend of results, in which no amount of hemicellulose was detected in Napier grass after pretreatet at 200 °C for 15 min. Both studies agreed that hydrothermal pretreatment plays a significant effect in improving the enzymatic hydrolysis yield afterwards. Meanwhile, Lee and Park [42] reported that sunflower biomass treated with hydrothermal pretreatment at 160–220 °C for 30 min demonstrated a reduction of hemicellulose and lignin up to 25 and 15%, respectively. This then led to higher methane yield (213.87–289.47 mL g^−1^) and biodegradability (43–63%) than the non-hydrothermally treated biomass. All of these reviews highlighted that despite of using a simple mechanism, hydrothermal can still efficiently removed impurities and improve the chemical and physical properties of lignocellulosic biomass prior to further treatments. 

#### 4.1.3. Steam Explosion

Steam explosion involves the use of high pressure and heat to pretreat lignocellulosic biomass. The biomass will be subjected to heat ranging from 160–280 °C and high pressure ranging from 0.2–5 MPa, depending on biomass source, duration, and other conditions [48,49]. Before the discovery of superheated steam and hydrothermal treatment, the steam explosion was widely applied in the industry due to its low energy consumption and chemical usage [49]. Figure 5 shows an example of the steam explosion process. Theoretically, biomass needs to be subjected to high temperature and pressure in a close reactor. The water contained in the biomass will then be evaporated and expanded, led to hydrolysis to a certain extent. Explosive decompression will then occur by promptly reducing the pressure to the atmospheric level [50,51]. 

Steam explosion treatment helps to reduce the particle size of biomass, disrupt the structure of lignocellulosic biomass by removing amorphous structures such as hemicellulose and other impurities, and reduce cellulose crystallinity [52]. Similar to hydrothermal, the steam explosion also carried out auto-hydrolysis. During processing, acetic acids and other organic acids will be formed, and this will assist in the breakdown of ester and ether bonds in the cellulose-hemicellulose-lignin matrix. For steam explosion, reaction temperature, pressure, and processing duration are considered as the key factors. 

Numerous studies have reported the effectiveness of this treatment in reducing impurities and enhancing the effectiveness of further treatments. For example, Abraham et al. [52] discovered that the sudden pressure drop due to explosion has pre-defibrillated the raw banana, jute, and pineapple leaf fibre biomass after pretreated for 1 h, which then eases and enhances the efficiency of fibrillation process by acid hydrolysis for the production of nanocellulose. Meanwhile, Medina et al. [51] discovered an application of steam explosion pretreatment for empty fruit bunches. The heating time was around 2 min and the reaction time was controlled after the temperature was reached. It was found that the application of steam explosion helped to enhance the production of glucans to 34.69%, reduce the amount of hemicellulose to 68.11%, and increase enzymatic digestibility to 33%. This was all due to steam explosion pretreatment, which helped in increasing the fibre porosity of empty fruit bunches. Marques et al. [53] also highlighted that the oil palm mesocarp fibre which has been treated to the steam explosion has higher purity, thermal stability, and crystallinity than the non-treated biomass. The reaction time was between 3 to 17 min. The cellulose pulp yield was increased by 47%. In addition, high-quality lignin was obtained as a co-product of steam explosion pretreatment, which can potentially be used for other purposes such as in the development of resin. 

### 4.2. Biological-Chemical Pretreatment

In recent years, a more often used combined pretreatment method is physical and chemical combined pretreatment, while biological and chemical combined pretreatment has yet to be thoroughly researched. Combining microbial and chemical pretreatments, for instance, is seen as a cost-effective technique for reducing pretreatment times, minimizing chemical usage and hence secondary pollution [54]. Table 3 listed different biological-chemical pretreatment approaches to pretreat lignocellulosic biomass. Till now, the biological-alkaline pretreatment for lignocellulosic biomass has been the most widely researched.

## 5. The Influence of Pretreatment of Natural Fibre on Several Applications 

Pretreatment of lignocellulosic materials has long been known for its advantages. It has been applied for various applications such as biocomposites, adsorbent, paper, packaging, military, biosugars, biomedical, bioenergy and more [55,56,57,58,59,60,61,62,63,64,65,66,67,68]. In Table 4, the purposes of the non-chemical pretreatment strategies and their benefits and drawbacks are summarised [69]. Since there are so many pretreatment-related applications, discussing each pretreatment technique in depth becomes very challenging. For certain applications, pretreatment techniques applied on natural fibre are summarised in the following sections.

### 5.1. Influence of Physical Pretreatment on Applications

Physical pretreatment is responsible for the changes in specific surface area, particle sizes, crystallinity index, or polymerization degree of biomass. The physical pretreatment avoids the use of chemicals, thus reducing the generation of waste and inhibitors for subsequent reactions. The management of biomass after harvesting, storage, and transportation is made easier by a higher bulk density [70]. Reduced particle size and increased specific surface area facilitate the following process by establishing a phase barrier between lignocellulosic material and chemicals and eliminating heat transfer limitation [71]. Mechanical, microwave or ultrasound pretreatments are the most common techniques carried out to improve the efficiency of the main steps in biomass processing.

It has been discovered that milling leads to higher production of biogas, bioethanol, and biohydrogen. Given the high energy requirements of industrial milling and the increasing energy demand, it seems doubtful that milling will be economically viable [69]. While most studies demonstrated that milling after chemical pretreatment reduces the amount of energy used and the cost of solid-liquid separation, the amount of mixing in pretreatment slurries and fermentation inhibitors are avoided [72]. Thus, understanding the characteristics of the feedstock is critical for making the best choice of technique and equipment for mechanical processing, and this should guarantee an adequate cost-effectiveness balance [73].

De la Rubia et al. [74] discovered that the excessive reduction in biomass may lower biofuel generation and impede methane synthesis during anaerobic digestion by the formation of inhibitory volatile fatty acids (VFA). When coupled with other pretreatment techniques, size reduction is more successful. The greatest biogas generation from rice straw was achieved via a combination of milling, grinding, and heating treatment. Milling is beneficial since it eliminates inhibitors of fermentation such as furfural and hydroxyl methyl furfural [75]. There have also been suggestions for other types of physical pretreatment, including the use of gamma rays to break the ß-1,4 glycosidic linkages, which results in a higher surface area and a lower crystallinity [76]. Ball milling pretreatment gave the lowest particle size compared to mashing or chipping but resulted in a lower hydrolysis rate [77]. On a wide scale, this technique will certainly be extremely costly, and it will raise significant environmental and safety issues. The use of a twin-screw extruder for methane production may reduce 50% of hemicellulose content. This concomitantly increases the fraction of soluble chemicals, e.g., carbohydrates, proteins, lipids, minerals, and vitamins, and rapidly converted to 15–21% more biogas by methanogenic microorganisms [78]. Table 5 summarises the applications that used physical pretreatments and their yield improvement and product properties.

### 5.2. Influence of Biological Pretreatment on Applications

The majority of pretreatment methods involve costly instruments or equipment that consumes a lot of energy, depending on the process. Biomass conversion in particular requires a large amount of energy for physical and thermochemical operations. Biological treatment with different kinds of rot fungus is being recommended more than ever as a low-energy delignification technique. The pretreatment is renowned for working with fungal and enzyme-assisted processes to break down the barrier that has formed within the cell wall, allowing for more abundant lignocellulosic components to be utilised in the activities of cellulase enzymes, hence increase their digestibility and yield. For instance, a pretreatment may enhance the enzymatic hydrolysis rate by 3–10-fold [1].

Additionally, any pretreatment should prevent carbohydrate degradation or loss, as well as the production of by-products that are detrimental to future hydrolysis and fermentation. The presence of white-rot fungus allows the organism to delignify, without compensating for the carbohydrate content, resulting in enhanced 30–35% cellulose conversion to sugar [95] and an additional 10–96% methane production [96,97]. In contrast to thermochemical techniques, chemical pretreatment suffers from silica scaling that prohibits the recovery of alkaline chemicals, due to the high silica concentration of many agricultural feedstocks, such as rice and wheat straw. The economic feasibility of scaling up biological pretreatment is higher since it does not need a large initial capital investment due to the lack of or reduced use of chemicals and heat, as well as the absence of a necessity for feedstock size reduction [19]. A further disadvantage of the thermochemical method is that it often produces low-molecular-mass molecules with high pretreatment severities, which may act as an inhibitor to the primary process [98]. As a result, it needs a detoxification step after the thermochemical reaction, which adds to the cost [99].

Another possibility of biological pretreatment is the potential to produce a variety of value-added co-products or intermediates, including enzymes, reducing sugars, furfural, ethanol, protein and amino acids, carbohydrates, lipids, organic acids, phenols, activated carbon, degradable plastic composites, cosmetics, adsorbents, resins, medicines, foods and feeds, methane, pesticides, promoters, secondary metabolites, surfactants, fertiliser, and other miscellaneous products [100,101,102,103,104]. Despite many successful attempts, economic separation and co-products recovery have remained a problem. Nonetheless, the diversity of the product allows for a wide range of markets, which means that market saturation is less of a concern [97].

Despite the advantage of requiring no additional nutrients, the usual fungal breakdown process needs a lengthy incubation period of up to 14–56 days [105]. Carbohydrates also gradually degrade over this period, which results, even with selective lignin-degrading fungus, in a reduced sugar yield, therefore making fungal biological pretreatment is impracticable for use in industrial production. The use of enzymes rather than fungus may overcome these significant drawbacks, including less carbohydrate consumption, shorter treatment time, and better yield [106]. However, only a limited number of enzyme treatments are as efficacious in pulping as fungal treatments, since solid wood enzymes cannot penetrate effectively and need high pressure to get better results [97]. Table 6 summarises the applications that used biological pretreatments and their yield improvement and product properties.

## 6. Challenges and Future Recommendations 

There is currently an issue with agro-industrial waste disposal across the world. Therefore, it is vital to continuously explore for alternatives to manage the problem effectively. A review of recent advances in the effect of different pretreatment techniques on the conversion of natural fibres to bioproducts has been discussed. It can be inferred that the downstream application has a profound effect on the selection and optimization of a feasible pretreatment technique. Among all of these, non-chemical approaches for natural fibres pretreatment are gaining popularity since they are more advantageous and greener than chemical pretreatment due to their chemical-free processability, cost-effectiveness, and sustainability. This is due to the fact that an ideal natural fibres pretreatment should have minimum or no solvent costs and also the capacity to process at high solids loadings with shorter treatment times and minimal inhibitor formation. 

In fact, each pretreatment has its own set of limitations or shortcomings, and no specific technique can be used to pretreat all types of biomasses. Hence, a thorough understanding of the relationship between biomass structure and pretreatment is needed. Each pretreatment has a substantial impact on fibre properties. The selection of pretreatment is determined by the widespread application of natural fibre materials. Several factors such as type of fibre, crystallinity, molecular weight and other properties may influence in selecting the most effective pretreatment method. Additionally, operating conditions, such as temperature, time, etc must be taken into consideration during pretreatments as they have a direct influence on the fibre properties. 

As mentioned previously, each pretreatment method has its benefits and shortcomings depending on the source of biomass, the processes employed, and the desired end product. Nevertheless, many previous studies have been conducted on a small scale, yet there is a significant disparity between laboratory preliminary findings, pilot-scale outcomes, and, eventually, industrial-scale results. Hence, further research is needed to address these issues and provide a feasible pretreatment approach for large-scale biorefinery operations. 

Besides that, utilization of by-products derived from the pretreatment is also important to be investigated. For example, the SHS pretreatment had partially degraded the lignocellulosic structure of the biomass into smaller compounds such as acetic acid, formic, levulinic and succinic. These compounds were found useful to be used as antimicrobial agents. This indicates the possibility of having lignocellulosic components degradation products as byproducts during SHS pretreatment [33]. However, to the best of our knowledge, lack of reports was focused on the other type of pretreatments. Moreover, it is also important to ensure that there is no consequence generation of contaminants could be derived during the pretreatment process of fibres. 

Even though most of the non-chemical pretreatment as discussed here are known to be more environmentally friendly, improvements are still needed. This is due to the fact that some of the techniques such as milling, SHS, and hydrothermal pretreatment require high production cost, especially at industrial levels. The high energy consumption and long processing time related to the pretreatment of fibres is still an issue that hampers the industrial applicability of some of these pretreatments. However, to the best of our knowledge, progress has been accomplished in this area, and numerous studies to overcome this issue have now been conducted worldwide. 

## Figures and Tables

**Figure 1 polymers-13-02971-f001:**
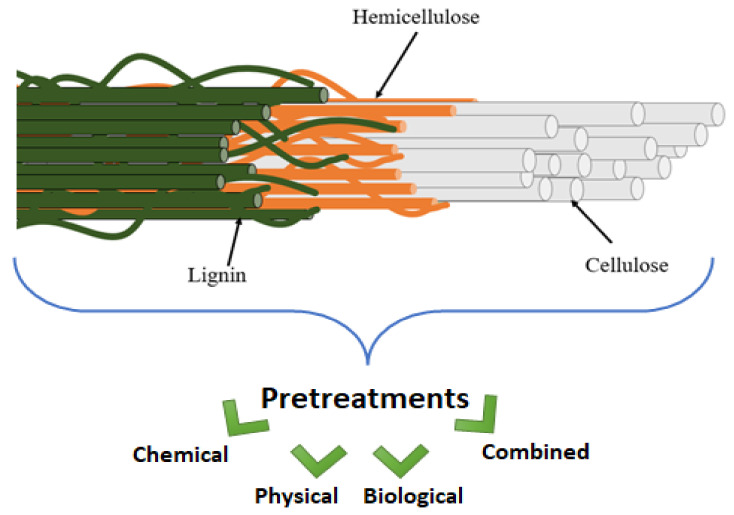
Overview of the complex structure of natural fibers and pretreatments.

**Figure 2 polymers-13-02971-f002:**
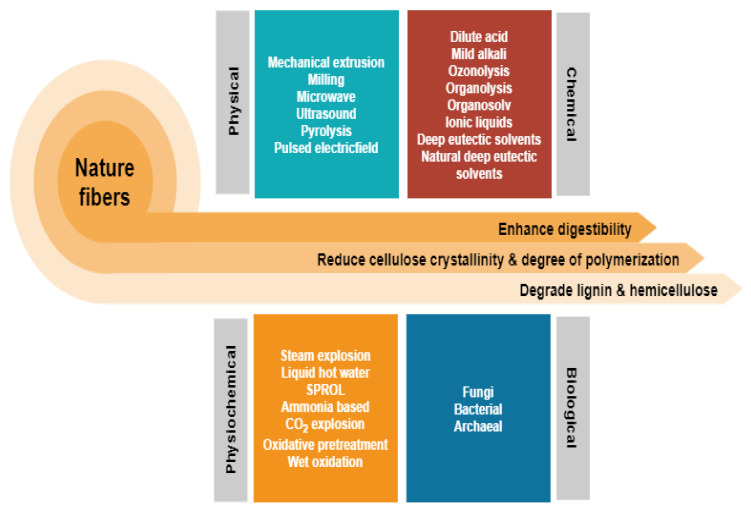
Different pretreatments, which fall into four main categories: physical, chemical, biological, and combination have been used to improve lignocellulosic fractionation for natural fibres.

**Figure 3 polymers-13-02971-f003:**
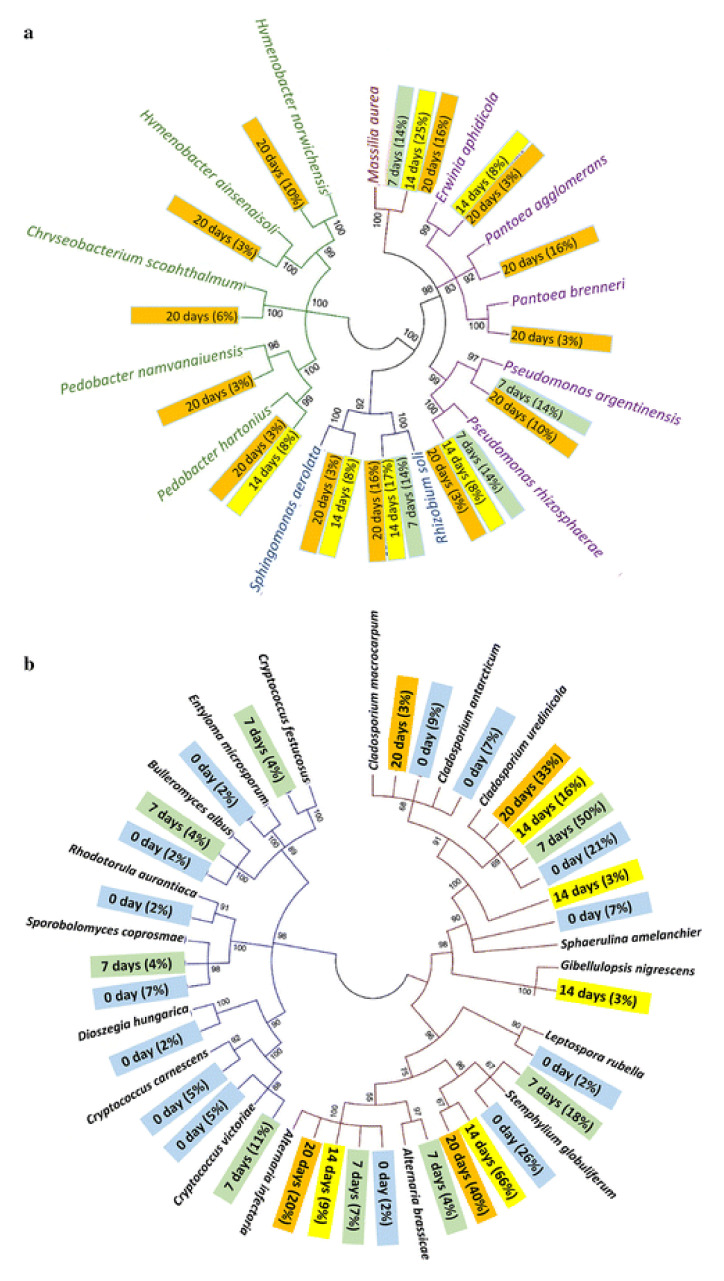
The phylogenetic tree of the (**a**) bacterial and (**b**) fungus communities found in hemp fibre samples. The color of the branches indicates the type of proteobacteria present, while the color of the tag indicates the number of bacteria/fungi present on different days [28].

**Figure 4 polymers-13-02971-f004:**
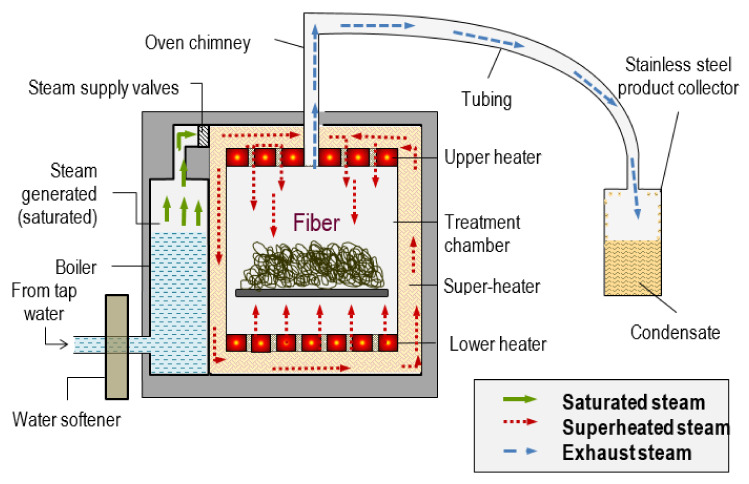
Schematic design of superheated steam pretreatment. Reprinted with permission from ref. [31]. 2018 Universiti Putra Malaysia.

**Figure 5 polymers-13-02971-f005:**
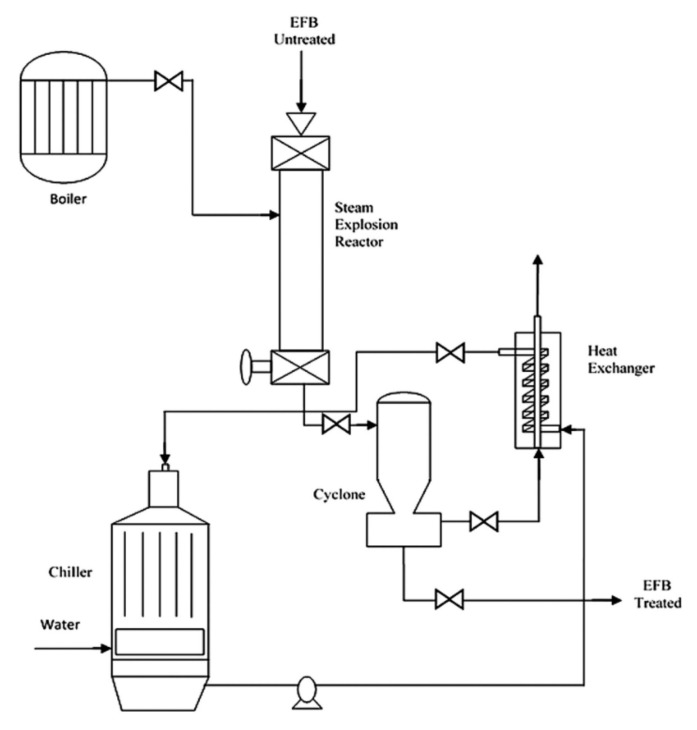
Schematic diagram of steam explosion process. Page: 11 Reprinted with permission from ref. [51]. 2016 Elsevier.

**Table 1 polymers-13-02971-t001:** Recent review articles related to greener pretreatment approaches for lignocellulosic biomass.

No.	Title	Highlights of Review	Ref.
1.	Enzymatic pretreatment of lignocellulosic biomass for enhanced biomethane production-A review	Reviews the anaerobic digestion process, challenges in degrading lignocellulosic materials, the current status of research to improve the biogas rate and yield from the anaerobic digestion of lignocellulosic biomass via enzymatic pretreatment, and the future trend in research for the reduction of enzymatic pretreatment cost	[12]
2.	A review on the environment-friendly emerging techniques for pretreatment of lignocellulosic biomass: Mechanistic insight and advancement	Discusses the important aspects of the emerging pretreatment techniques of lignocellulosic biomass including the advancements, and the mechanistic insight for large scale of commercial implementation in a lignocellulosic biorefinery.	[13]
3.	Recent Insights into Lignocellulosic Biomass Pyrolysis: A Critical Review on Pretreatment, Characterization, and Products Upgrading	Provides an outline of the pyrolysis process including physical and chemical pretreatment of biomass, pyrolysis mechanism, and process products upgrading.	[14]
4.	Recent advances in the pretreatment of lignocellulosic biomass for biofuels and value-added products	Briefly presents recent findings on the chemical pretreatment for the conversion of lignocellulosic materials into fuel and value-added products.	[15]
5.	Emerging technologies for the pretreatment of lignocellulosic biomass	Reviews the application of emerging technologies in chemical and mechanical pretreatment.	[16]

**Table 2 polymers-13-02971-t002:** Highlights of ultrastructural changes on hemp stems and fibres associated with microbial activity during the retting process [29].

Retting Period	0 Days	7 Days	14–20 Days	After 50 Days
Changes in the hemp stem’s and fibre’s ultrastructure	(i) Stem with a well-preserved layered structure (ii) Un-collapsed, unbroken cells with their original cell geometry (iii) Living cells with cytoplasm (iv) Cuticle and trichomes are unharmed on the clear surface. (v) Chloroplasts in abundance in the upper epidermis	(i) The structure as a whole is in good condition. (ii) Fungal growth on the outside of the stems and inside the stems (iii) With damaged epidermis and parenchyma, cellular architecture is less stable.	(i) Cuticle has seriously deteriorated. (ii) Changes in cellular anatomy, as well as significant loss of live cells (iii) Fibre bundles were isolated from each other and the epidermis. (iv) Thick-walled cells populate seldom; parenchyma degrades completely, although chlorenchyma suffers less harm. (v) Bast fibres with sporadic moderate attacks (vi) Fungi colonisation and decay morphology were both affected by fibre morphology.	(i) The structure of hemp was severely harmed and dissolved. (ii) The epidermis and cambium were heavily invaded by dominating bacteria. (iii) In the bast regions, the parenchyma cells have been destroyed, and the structural integrity has been lost. (iv) All cell types, including fibre cells, have hyphae inside their lumina. (v) BFIs are more intense inside the stem. (vi) Anatomy and ultrastructure have been severely harmed. (vii) Bast fibres with a thick wall and degradation properties (viii) Effects on the ultrastructure of the fibre wall. CML loosening/degradation, resulting in delamination and defibration The S3 layer is loosening and decaying Delamination within the S2 trans wall and intra wall cracks in the S2 layer have a noticeable effect S2 materials have been removed directly (e.g., S2 thinning, broken S2, and disintegration into nanosized cellulose fibrillar structures)
The dynamics and activity of microbes	Fungi (i) Rarely seen Bacteria (ii) Not observed Fungi	Fungi (i) Mycelia with sparse growth (ii) Less variety (iii) Outside of the cortical layers, colonisation occurs largely in live cells. (iv) Trichomes near to the surface trichomes have dense colonisation. (v) Dependence on readily available food (vi) Damage to cell walls is reduced. Bacteria (i) Less abundant	Fungi (i) Extensive and plentiful (ii) Mycelia densely covering the cuticle (iii) diverse population (iv) a large number of spores (v) Interactions and activities that are intense Bacteria (i) Abundant (ii) Diverse population iii) Over the cuticle, colonies (iv) Associated with hyphae and fungal spores (v) After 20 days, there are more noticeable activity (vi) Cuticle has severely deteriorated	Fungi (i) Less abundant on the outside of the stem (ii) Mycelia on the surface is dead, but there are active hyphae inside the stem (iii) Mycelia, an invading bacteria’s sole source of nourishment, showed bacterial mycophagy (i.e., extracellular and endocellular biotrophic and extracellular necrotrophic activities). Bacteria (i) Highly abundant inside and outside the stems (ii) Highly dominant and diverse role. (iii) Visible as dense overlay representing (a) Biofilms (b) Morphologically different colonies (c) Randomly scattered cells (iv) Showed strong BFIs (v) Using fungal highways, bacterial movement occurs over and inside the hemp stem. (vi) Cutinolytic and cellulolytic activities were improved.

**Table 3 polymers-13-02971-t003:** Previous research on biological-chemical pretreatment approaches to pretreat lignocellulosic biomass. Data retrieved from Ref. [54].

Substrate	Conditions		Component’s Degradation (%)		
	1st Step	2nd Step	Lignin	Hemicellulose	Cellulose
Biological—alkaline pretreatment					
Corn stalks	*Irpex lacteus* (28 °C, 15 d)	0.25 M NaOH solution (75 °C, 2 h)	80	51.37	6.62
*Populus tomentosa*	*Trametes velutina* D10149 (28 °C, 28 d)	70% (*v*/*v*) ethanol aqueous solution containing 1%(*w*/*v*) NaOH (75 °C, 3 h)	23.08	22.22	18.91
Willow sawdust	*Leiotrametes menziesii* (27 °C, 30 d)	1% (*w*/*v*) NaOH (80 °C, 24 h)	59.8	68.1	51.2
	*Abortiporus biennis* (27 °C, 30 d)		54.2	51.8	29.1
Biological—acid pretreatment					
*Populus tomentosa*	*Trametes velutina* D1014 (28 °C, 56 d)	1% sulphuric acid (140 °C, 1 h)	23.82	75.96	(+) 18.74
Oil palm empty fruit bunches	*Pleurotus floridanus* LIPIMC996 (31 °C, 28 d)	Ball milled at 29.6/s for 4 min. Phosphoric acid treatment (50 °C, 5 h)	(+) 8.29	60.63	(+) 37.52
Olive tree biomass	*Irpex lacteus* (Fr.238 617/93) (30 °C, 28 d)	2% *w*/*v* H_2_SO_4_ (130 °C, 1.5 h)	(+) 105.82	75.29	(+) 62.95
Biological—oxidative pretreatment					
Corn Straw	*Echinodontium taxodii* (25 °C, 15 d)	0.0016% NaOH and 3% H_2_O_2_ (25 °C, 16 h)	52.00	23.64	(+) 45.45
Hemp chips	*Pleurotus eryngii* (28 °C, 21 d)	3% NaOH and 3% (*v*/*v*) H_2_O_2_ (40 °C, 24 h)	55.7	23.2	25.1
Biological—organosolv pretreatment					
Sugarcane straw	*Ceriporiopsis subvermispora* (27 °C, 15 d)	Acetosolv pulping (Acetic acid with 0.3% *w*/*w* HCl) (120 °C, 5 h	86.8	93.8	32.1
*Pinus radiata*	*Gloeophyllum trabeum* (27 °C, 28 d)	60% ethanol in water solvent (200 °C, 1 h)	74.26	80.74	-
Biological—liquid hot water (LHW) pretreatment					
Soybean	Liquid Hot water (170 °C, 3 min, 400 rpm, 110 psi, solid to liquid ratio of 1:10)	*Ceriporiopsis subvermispora* (28 °C, 18 d)	36.69	41.34	0.84
Corn stover			41.99	42.91	7.09
Wheat straw	Hot water extraction (HWE) (85 °C, 10 min, solid to liquid ratio of 1:20)	*Ceriporiopsis subvermispora* (28 °C, 18 d)	24.87	13.19	1.86
Corn stover			30.09	28.14	4.96
Soybean			0.09	0.09	0.09
Biological—steam explosion pretreatment					
Beech woodmeal	*Phanerochaete chrysosporium* (37 °C, 28 d)	Steam explosion (215 °C, 6.5 min)	42.00	-	-
Sawtooth oak, corn and bran	*Lentinula edodes* (120 d)	Steam explosion (214 °C, 5 min, 20 atm)	17.1	80.43	(+) 5.19

(+): represents the increment in fibre content.

**Table 4 polymers-13-02971-t004:** Purposes of the pretreatment strategies and their advantages and disadvantages.

Pretreatments	Preferred Natural Fibres	Purposes	Advantages	Disadvantages
Physical	Hardwoods and agricultural residues	Enhance the digestibility of lignocellulosic biomass by increase the available specific surface area, and reduce both the degree of polymerisation and cellulose crystallinity	(1) No recycling cost (2) No chemical usage (3) Increase biogas, bioethanol and biohydrogen yields	(1) Excessive size reduction decreases biofuel production (2) Formation of fermentation inhibitors at high temperature (3) Incomplete digestion of lignin-carbohydrate matrix (4) The need to wash the hydrolysate decreases sugar yield (5) High energy requirement
Biological	Hardwoods, softwoods, and agricultural residues	Leverage the action of fungi capable of producing enzymes that can degrade lignin, hemicellulose, and polyphenols	(1) The depolymerisation is very selective and efficient (2) Low-capital cost (3) Low energy requirement (4) No chemicals requirement (5) Mild process conditions	(1) The rate of biological pretreatment is too slow for industrial purposes (10–14 days) (2) Require careful growth conditions and a large amount of space (3) A fraction of carbohydrate is consumed by the microbes, thus reduces the sugar yield

**Table 5 polymers-13-02971-t005:** A summary of physical pretreatments applied for numerous applications.

Bioproducts	Natural Fibres	Pretreatments	Conditions	Yield Improvement/Product Properties	References
Biohydrogen	Corn stover	Steam explosion	1.5 Mpa and 198 °C for 1.5 min	51.9 L H_2_ kg^−1^ TS *	[79]
	Rice straw	Hydrothermal	pH 7.0, 210 °C, 15.4 °C min^−1^, and 20% TS	28.0 mL H_2_ g^−1^ VS *	[80]
Biomethane	Sugarcane bagasse	Hydrolysis	178.6 °C, 43.4 min, and solid to liquid ratio of 0.24	1.56 Nm^3^ CH_4_ kg^−1^ TOC *	[81]
	Wheat straw	Microwave irradiation	260 °C, 33 bars, 3 min	28%	[82]
	*Pennisetum* hybrid			12%	[83]
	Blend of maize, ryegrass, and rice straw	Extrusion	Exit slit opened at 60%	11.5–13.4%	[84]
	Hay	Steam explosion	220 °C for 15 min	16%	[85]
	Vine trimming shoot	Extrusion	200 g h^−1^ feed rate	51–58% hemicellulose reduction, 15.7–21.4% CH_4_ increased	[78]
Biosugar	Wheat straw	Supercritical CO_2_ & steam explosion	A steam explosion at 200 °C for 15 min and supercritical CO_2_ of 12 MPa at 190 °C for 60 min	36.5%	[86]
	Poplar wood chips	Mechanical pulping & steam	Disc clearance set 0.5–0.1 mm for mechanical pulping and steam pretreatment at 210 °C for 5 min	76%	[87]
	Poplar wood	Steam explosion	180 °C and 18 min	94%	[88]
	Cane bagasse	Hydrothermal	200 °C	4 mg xylose ml^−1^ *	[89]
	Pinewood		240 °C and 10 min	32% **	[90]
	Rapeseed meal		260 °C and 10 min	51 g glucose kg^−1^ *	[91]
Nanocellulose	Poplar wood	Steam explosion	2 MPa for 180 s	13.2%	[92]
	Cotton	High-pressure homogenization	80 MPa for 30 HPH	10–20 nm in diameter, reduced thermal stability, and crystallinity	[93]
	Sugarcane bagasse			10–20 nm in diameter, reduced thermal stability, and crystallinity	[94]
	Oil palm biomass	Superheated steam	260 °C for 30 min	<100 nm diameter, 27% crystallinity reduced	[35]

* The highest yield obtained. ** In carbohydrate.

**Table 6 polymers-13-02971-t006:** A summary of biological pretreatments applied for numerous applications.

Bioproducts	Natural Fibres	Type of Microbes/Enzymes	Hydrolysis Conditions	Yield Improvement/Product Properties	References
Biohydrogen	Corn stover	*Clostridium cellulolyticum* and hydrogen fermentation bacteria	20 mL of medium, 5% (*v*/*v*) inoculum, 10 g L^−1^ carbon source, at 37 °C for 96 hrs	40.3 L H_2_ kg^−1^ TS *	[79]
Bioethanol	Corn stover	*Ceriporiopsis subvermispora*	28 °C for 18 days	57.8% yield increased	[107]
	Corn stover	*Ceriporiopsis subvermispora*	28 °C for 35 days	66.6% yield increased	[107]
	Potato and cassava peel	*Gloeophyllum sepiarium* and *Pleurotus ostreatus*	28 °C for 7 days	26% yield increased	[108]
	Straw	*Neosartorya fischeri–Myceliophthora thermophila* and *Aeromonas hydrophila*–*Pseudomonas poae*	30–55 °C for 6 days	7-fold yield increased	[109]
	Corn stover	*Irpex lacteus*	28 °C for 42 days	66.9% yield increased	[110]
	Corn stalks		28 °C for 28 days	82% yield increased	[111]
Biomethane	Wheat straw	*Trametes versicolor*	Laccase at 500 U/L, 25 °C for 6 days	10–18% yield increase	[96]
	Cassava	Yeast and cellulolytic bacteria	100 mL of PCS medium, at 55 °C for 12 h	96.6% yield increased	[112]
	Microalgae	Enzyme mix (cellulase, glucohydrolase and xylanase)	1% enzyme mix, 37 °C for 24 hrs	15% yield increased	[113]
	Sawdust	*Methanobrevibacter thaueri* MB-1, *Methanosarcina acetivorans* MB-2, and *Methanococcus voltae* MB 3.	60 °C for 6 days	92.2% yield increased	[114]
Biosugar	Corn stover	*Ceriporiopsis subvermispora*	28 °C for 5–7 days	57–67% yield increase	[107]
	Silver grass	*Bacillus, Pseudomonas, Exiguobacterium*, and *Aeromonas*	37 °C for 7 days	2.2-fold yield increased	[115]
	Sugarcane bagasse	*Ceriporiopsis submervispora*	27 °C for 60 days	47% yield increased	[116]
	Sawdust	*Pleurotus pulmonarius*	28 °C for 30 days	94.8% yield increased	[117]
	Paddy straw	*Pleurotus florida*	25–29 °C for 28 days	75.3% yield increased	[118]
	Rice straw	*Pholiota adiposa*	25 °C for 120 h	716 mg g^−1^ *	[119]
		*Pholiota adipose* and *Armillaria gemina*	27 °C for 45 days	74.2% yield increased	[120]
	*Populus tomentiglandulosa*	*Armillaria gemina* SKU2114	30 °C for 48 h	62% yield increased	[121]
Nanocellulose	Eucalyptus	Endoglucanase and cellobiohydrolase	7 pH, 50 °C for 48 h	20 nm diameter, >500 nm length	[122]
	Wood fibre	Endoglucanase	4.8 pH, 50 °C for 2 h	5–30 nm diameter	[123]
	Orange residues	ß-glucosidase	4 pH, 50 °C for 48 h	180 nm diameter, 1.3 mm length	[124]
	Sugarcane bagasse	ß-glucosidase and endoglucanase	5 pH, 50 °C for 24 h	14–18 nm diameter, 195–250 nm length	[125]
	Maple pulp	Cellic CTec 2 and Cellic HTec 2 (commercial enzymes)	4.8 pH, 50 °C for 72 h	5–10 nm diameter, 1 μm length	[126]
	Cotton linters	Cellulase	5 pH, 55 °C for 24 h	35 nm diameter, 0.3 mm length	[127]

* The highest yield obtained.

## Data Availability

Not applicable.

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
