# Peer review of "Greener Pretreatment Approaches for the Valorisation of Natural Fibre Biomass into Bioproducts"

_polymers, 2021, doi:10.3390/polym13172971_

Round 1
Reviewer 1 Report
The manuscript needs formatting corrections (see the corrections and comments in the attached file). The manuscript presents an extensive bibliographic review with current references.

Author Response
Response to reviewer’s comments
Reviewer 1
Comment 1
Page 1
Correction:
Many industries currently produce many tons of agro-industrial wastes, however, direct utilization of lignocellulosic biomass as a feedstock for bioproducts is challenging due to their complex structure as represented in Figure 1. A variety of useful components, including sugars sugar, protein, lipids, cellulose, and lignin, are present in the natural fibers...
Response: Thank you for your comment. We had revised the sentence. (Page 1)
Comment 2
Page 2
Suggestion: Figure 1. Overview of natural fibers pretreatment.
Change to:
Figure 1. Overview of the complex structure of natural fibers and pretreatments
Response: Thank you for your comment. We had revised the title for Figure 1. (Page 2)
Comment 3
Page 4
Temperature and screw speed of extrusion are the main important factors. [11]. Studied the effect these factors on pretreatment of corn cobs. When pretreatment…
Suggestion:
Temperature and screw speed of extrusion are the main important factors. Karunanithy and Muthukumarappan [11] studied the effect these factors on pretreatment of corn cobs. When pretreatment…
OR
Studies have been carried out on the effect these factors on pretreatment of corn cobs [11]. When pretreatment…
Response: Thank you for your comment. We had revised the sentence. (Page 4)
Comment 4
Page 4
Wet disk milling has been a popular mechanical pretreatment because of its low energy consumption as compared to other milling processes. Disk milling enhances cellulose hydrolysis by producing fibers and more effective as compared to hammer milling which produces finer bundles [12]. [13] compared the effect of wet disk milling and conventional ball milling pretreatment method over rice straw.
Suggestion:
Wet disk milling has been a popular mechanical pretreatment because of its low energy consumption as compared to other milling processes. Disk milling enhances cellulose hydrolysis by producing fibers and more effective as compared to hammer milling which produces finer bundles [12]. Hideno et al. [13] (2009) compared the effect of wet disk milling and conventional ball milling pretreatment method over rice straw.
Response: Thank you for your comment. We had revised the sentence. (Page 5)
Comment 5
Page 5
Correction:
Superheated steam has been managed to alter the chemical composition of natural fibers. It has been proven that superheated steam pretreatment managed to remove high amount of hemicellulose from the lignocellulose fibers [18–21 22]. According to Warid et al. (2016) [22], superheated Superheated steam pretreatment on oil palm biomass at higher temperature and shorter time managed to remove high amount of hemicellulose while maintaining the cellulose composition as compared to the method reported by Norrrahim et al. (2018) [21]. It was…
Response: Thank you for your comment. We had revised the sentence. (Page 10)
Comment 6
Page 6
Hydrothermal …
hydronium ions (H+) generated from water and acetic groups released from hemicellulose. The hydronium ions (H+) will act as…
Sugestion: ...hydronium ions (H3O+)
Response: Thank you for your comment. We had revised the sentence. (Page 11)
Comment 7
NOTE: in this paragraph below, inform the time of the hydrothermal pre-treatment used by the cited authors.
Numerous studies have reported the effectiveness of this treatment in reducing im-purities, especially at a very high temperature. Zhang et al. (2018) [28] studied the effects of different hydrothermal temperatures which were 170, 190 and 210 °C on corn stover. This study reported a drastic reduction in hemicellulose with an increase in hydrothermal temperature. In fact, no content of hemicellulose was detected and almost 125% of lignin was removed after hydrothermal treatment at 210 °C. Similarly, Phuttaro et al. (2019) [29] also re-ported the same trend of results, in which no amount of hemicellulose was detected in Napier grass at 200 °C. Both studies agreed that hydrothermal pretreatment plays a sig-nificant effect in improving the enzymatic hydrolysis yield afterward. Meanwhile, Lee and Park (2020) [24] reported that sunflower biomass treated with hydrothermal pretreatment at 160 – 220 °C was having a reduction of hemicellulose and lignin up to 25 and 15%, respectively. This then led to higher methane yield (213.87 - 289.47 mL g-1) and biodegra-dability (43 – 63%) than the non-hydrothermally treated biomass. All these reviews high-lighted that despite of using a simple mechanism, hydrothermal can still efficiently re-moved impurities and improve the chemical and physical properties of lignocellulosic biomass prior to further treatments.
Numerous studies have reported the effectiveness of this treatment in reducing im-purities and enhancing the effectiveness of further treatments. For example, Abraham et al. [34] (2011) discovered that the sudden pressure drop due to explosion has pre-defibrillated the raw banana, jute, and pineapple leaf fibre biomass, which then eases and enhances the efficiency of fibrillation process by acid hydrolysis for the production of nanocellulose. Meanwhile, Medina et al. [33] (2016) discovered that the application of steam explosion helped to enhance the production of glucans to 34.69%, reduce the amount of hemicellulose to 68.11% and increase enzymatic digestibility to 33%. This was all due to steam explosion pretreatment, which helped in increasing fiber porosity of empty fruit bunches. Marques et al. [35] (2020) also highlighted that the oil palm mesocarp fiber which has been treated to the steam explosion has higher purity, thermal stability, and crystallinity than the non-treated biomass. The cellulose pulp yield was increased by 47%. In addition, high-quality lignin was obtained as a co-product of steam explosion pretreatment, which can poten-tially be used for other purpose such as in the development of resin.
Response: Thank you for your comment. We had revised the sentence by including the treatment time as requested. (Page 11 and Page 12)
Comment 8
Page 10
Table 1. Highlights of ultrastructural changes on hemp stems and fibers associated with microbial activity during the retting process [65]. ??? numering
NOTE: Following the order of numbering the references, the number would be 42 instead of 65.
Response: Thank you for your comment. We had followed the numbering order of the reference. (Page 8)
Comment 9
Page 14
Review the order of numbering of references From [63] of the text goes to [83] in Table 1.
Response: Thank you for your comment. We had reviewed the order of the references (Page 8)
Comment 10
References
NOTE: Reference 64 is not cited in the text.
Response: Thank you for your comment. We had updated all the references of the revised manuscript.
Reviewer 2 Report
The Authors proposed a review paper on various approaches towards lignocellulosic biomass pretreatment with a focus on green and sustainable methods. The review is comrehensive, however below I list several comments that may help to improve the quality of the paper:
- Introduction - please provide a list of review papers on lignocellulosic biomass pretreatment from last e.g. 5 years and underline clearly what is new in the proposed manuscript (how is it different from other review papers in the field).
- The Authors discussed physical, physicochemical and biological methods of biomass pretreatment, assuming those as "green" approaches. What about combined methods, even including chemical-biological approach? Maybe a small paragraph on combined methods would be reasonable in this review?
- Section "challenges and future recommendations" should be more informative. In my opinion, more precise summary of the previous discussions, however short, should be provided. Moreover, challenges should be clearly depicted as well as at least one more recommendation, beside scaling-up, should be proposed.
- English / language corrections: There are several similar language mistakes found in the manuscript. Please verify the English language according to some examples given below: Abstract, last sentence: pretreatment approach for the natural fibers; Figure caption - Fig. 1: Overview of nautral fibers pretreatment methods; Page 2, one line below Fig. 1: biomass hydrolysis; Physical pretreatment, 3rd sentence: remove "is"; Physical pretreatment lacks ability; pretreatment requires high energy; Milling: Paper [13]...; Ultrasouond: ...Duration of ultrasound pretreatment; Biological pretreatment, 3rd sentence: bacterium commonly found - remove "that".
- Why lines numeration is missing in the article template?
- Please correct the numeration/spacing, e.g. 2.2., 2.3., 3.2..
Author Response
Response to reviewer’s comments
Reviewer 2
Comment 1
Introduction - please provide a list of review papers on lignocellulosic biomass pretreatment from last e.g. 5 years and underline clearly what is new in the proposed manuscript (how is it different from other review papers in the field).
Response: Thank you for your comment. We had improved the introduction section by include several lists of review paper on lignocellulosic biomass pretreatment as presented in Table 1. We also revised some sentences in this section. (Page 3)
Comment 2
The Authors discussed physical, physicochemical and biological methods of biomass pretreatment, assuming those as "green" approaches. What about combined methods, even including chemical-biological approach? Maybe a small paragraph on combined methods would be reasonable in this review?
Response: Thank you for your comment. We had included the section of “combination pretreatment”(Page 9-Page 13)
Comment 3
Section "challenges and future recommendations" should be more informative. In my opinion, more precise summary of the previous discussions, however short, should be provided. Moreover, challenges should be clearly depicted as well as at least one more recommendation, beside scaling-up, should be proposed.
Response: Thank you for your comment. We had revised the section of “Challenges and future recommendations". (Page 18)
Comment 4
English / language corrections: There are several similar language mistakes found in the manuscript. Please verify the English language according to some examples given below: Abstract, last sentence: pretreatment approach for the natural fibers; Figure caption - Fig. 1: Overview of nautral fibers pretreatment methods; Page 2, one line below Fig. 1: biomass hydrolysis; Physical pretreatment, 3rd sentence: remove "is"; Physical pretreatment lacks ability; pretreatment requires high energy; Milling: Paper [13]...; Ultrasouond: ...Duration of ultrasound pretreatment; Biological pretreatment, 3rd sentence: bacterium commonly found - remove "that".
Response: Thank you for your comment. We had improved the English language as suggested.
Comment 5
Why lines numeration is missing in the article template?
Response: Thank you for your comment. The editorial teams had transferred the manuscript into the template. It does not have the lines numeration.
Comment 6
Please correct the numeration/spacing, e.g. 2.2., 2.3., 3.2..
Response: Thank you for your comment. The numeration and spacing were corrected.